# Preparation of Poly(L-lactide)-*b*-poly(ethylene glycol)-*b*-poly(L-lactide)/Zinc Oxide Nanocomposite Bioplastics for Potential Use as Flexible and Antibacterial Food Packaging

**DOI:** 10.3390/polym16121660

**Published:** 2024-06-11

**Authors:** Yaowalak Srisuwan, Prasong Srihanam, Surachai Rattanasuk, Yodthong Baimark

**Affiliations:** 1Biodegradable Polymers Research Unit, Department of Chemistry and Centre of Excellence for Innovation in Chemistry, Faculty of Science, Mahasarakham University, Kantarawichai 44150, Mahasarakham, Thailand; yaowalak.s@msu.ac.th (Y.S.); prasong.s@msu.ac.th (P.S.); 2Major of Biology, Department of Science and Technology, Faculty of Liberal Arts and Science, Roi-Et Rajabhat University, Selaphum 45120, Roi-Et, Thailand; surachai_med@hotmail.com

**Keywords:** poly(lactic acid), poly(ethylene glycol), block copolymer, zinc oxide, nanocomposites, antibacterial activity

## Abstract

High-molecular-weight poly(L-lactide)-*b*-poly(ethylene glycol)-*b*-poly(L-lactide) (PLLA-PEG-PLLA) is a flexible and biodegradable bioplastic that has promising potential in flexible food packaging but it has no antibacterial ability. Thus, in this work, the effect of zinc oxide nanoparticles (nano-ZnOs) which have antimicrobial activity on various properties of PLLA-PEG-PLLA was determined. The addition of nano-ZnOs enhanced the crystallization, tensile, UV-barrier, and antibacterial properties of PLLA-PEG-PLLA. However, the crystallization and tensile properties of nanocomposite films decreased again as the nano-ZnO increased beyond 2 wt%. The nano-ZnO was well distributed in the PLLA-PEG-PLLA matrix when the nano-ZnO content did not exceed 2 wt% and exhibited some nano-ZnO agglomerates when the nano-ZnO content was higher than 2 wt%. The thermal stability and moisture uptake of the PLLA-PEG-PLLA matrix decreased and the film’s opacity increased as the nano-ZnO content increased. The PLLA-PEG-PLLA/ZnO nanocomposite films showed good antibacterial activity against bacteria such as *Escherichia coli* and *Staphylococcus aureus*. It can be concluded that nano-ZnOs can be used as a multi-functional filler of the flexible PLLA-PEG-PLLA. As a result, the addition of nano-ZnOs as a nucleating, reinforcing, UV-screening, and antibacterial agent in the flexible PLLA-PEG-PLLA matrix may provide protection for both the food and the packaging during transportation and storage.

## 1. Introduction

In the past few decades, biodegradable plastics have been extensively investigated for use as food packaging in place of non-biodegradable petroleum-based plastics because they might circumvent the serious problems of plastic waste. Furthermore, biodegradable bio-based plastics, also known as biodegradable bioplastics, have received a lot of attention for use as food packaging because of their environmentally friendly properties such as biocompatibility, renewability, and sustainability, etc. [1,2,3]. Among biodegradable bioplastics, poly(L-lactic acid) or poly(L-lactide) (PLLA) synthesized from bio-derived L-lactic acid is the one that has received the most attention. The reason for this is that PLLA has good thermal processability [3,4,5], good mechanical properties [6], and market availability [7]. Therefore, PLLA demonstrates acceptable qualities for food packaging that involves food contact.

For food packaging, maintaining the freshness and quality of the food during transportation and storage is important, and will extend the shelf life of food and also reduce food waste [3,8,9]. Thus, antimicrobial packaging was developed in order to extend the shelf life storage of food and reduce food waste [1,3]. PLLA-based materials with antimicrobial ability for use in food packaging applications have been prepared by incorporating various antimicrobial agents [1].

The antimicrobial agents can be classified into two classes: (1) including natural antimicrobial agents such as chitosan [10] and plant-derived essential oils [11], etc., and (2) synthetic antimicrobial agents such as silver nanoparticles (nano-Ags) [12], zinc oxide nanoparticles (nano-ZnOs) [13,14], and titanium dioxide nanoparticles (nano-TiO_2_s) [14], etc. Nowadays, the nano-ZnO has received widespread interest in potential applications as an antimicrobial agent in food packaging because it is non-toxic, low cost, and available at commercial scale, and it has high thermal stability, good UV absorption, and strong antimicrobial activity [13,14,15,16]. Moreover, the United States Food and Drug Administration (FDA, 21CFR182.8991) has classified the nano-ZnO as a generally recognized as safe (GRAS) substance, and the European Union has approved it for use in certain food contact applications [15,17]. The nano-ZnO has been widely used to foster the antimicrobial properties of both the non-biodegradable plastics such as poly(ethylene terephthalate) (PET) [18], linear low density polyethylene (LLDPE) [19], polystyrene (PS) [20], and polypropylene (PP) [21] as well as the biodegradable plastics such as chitosan [22], pectin [15], poly(3-hydroxybutyrate) (PHB) [23], poly(butylene adipate-*co*-terephthalate) (PBAT) [24], and PLLA [15,25,26,27]. PLLA/ZnO nanocomposites have also been studied for use in biomedical fields that require its good biocompatibility [16]. Thus, the PLLA/ZnO nanocomposites are safe for use in food packaging.

However, a major disadvantage of PLLA that limits its widespread use is its low flexibility due to its high glass-transition-temperature (*T_g_* about 60 °C) [28]. High-molecular-weight PLLA-*b*-poly(ethylene glycol)-*b*-PLLA (PLLA-PEG-PLLA) triblock copolymers exhibited greater flexibility than the PLLA [28,29]. The PEG middle-blocks acted as plasticizing sites to decrease the *T_g_* of PLLA end-blocks. The PEG chains are biocompatible and biodegradable and are FDA-approved materials [30]. Moreover, PLLA-PEG-PLLA showed a faster biodegradation rate than PLLA due to the hydrophilicity of PEG middle-blocks [31]. Therefore, the biodegradable PLLA-PEG-PLLA has great potential as single-use food packaging materials with excellent flexibility. However, PLLA-PEG-PLLA/ZnO nanocomposites with antimicrobial properties have not been prepared and reported so far.

In this study, we propose flexible and antibacterial PLLA-PEG-PLLA, in which are incorporated nano-ZnOs as antimicrobial agents. The PLLA-PEG-PLLA nanocomposites containing nano-ZnOs in the range of 1 wt% to 4 wt% were prepared for the first time by a melt blending method. The effect of the nano-ZnO content on crystallization, thermal decomposition, morphological, mechanical, UV-barrier, and antibacterial properties of the PLLA-PEG-PLLA were investigated and compared with the pure PLLA-PEG-PLLA. In particular, the goal of this work was to prepare flexible and biodegradable materials for use in food packaging with improved crystallization, mechanical, UV-barrier, and antibacterial properties.

## 2. Materials and Methods

### 2.1. Materials

High-molecular-weight PLLA-PEG-PLLA was synthesized through in situ chain extension and ring-opening polymerization of L-lactide monomer in the presence of 2.0 phr chain extender at 165 °C for 6 h, as previously described by us [32,33]. PEG with molecular weight of 20,000 and stannous octoate were used as the initiating system. Joncryl^TM^ ADR-4368 (BASF, Bangkok, Thailand) was used as a chain extender. The number-averaged molecular weight (*M_n_*) and dispersity (Ð) obtained from gel permeation chromatography (GPC) of the PLLA-PEG-PLLA were 108,500 and 2.2, respectively [33]. ZnO nanoparticles (nano-ZnOs) with purity of 99.5% and averaged particle size of 33 nm were purchased from Nanoscience Technology Co., Ltd. (Bangkok, Thailand) and used without further purification. Figure 1 shows an SEM image of ZnO nanoparticles.

### 2.2. Preparation of PLLA-PEG-PLLA/ZnO Nanocomposites

PLLA-PEG-PLLA/ZnO nanocomposites were prepared by melt blending using an internal mixer (HAAKE, Polylab OS System, Waltham, MA, USA) at 190 °C for 10 min with a rotor speed of 100 rpm. Both the PLLA-PEG-PLLA and the nano-ZnO were dried in a vacuum oven at 50 °C overnight to remove residue moisture before melt blending. Nano-ZnO contents of 1 wt%, 2 wt%, 3 wt%, and 4 wt% were investigated. Nanocomposite films with 0.2–0.3 mm thickness were prepared by compression molding using a compression molding machine (Carver, Auto CH, Wabash, IN, USA). The nanocomposites were dried in a vacuum oven at 50 °C overnight to remove residual moisture before compression molding. The nanocomposites were heated at 190 °C for 3 min without compression force before compressing under 5 MPa load for 3 min. The resulting films were then cooled with water-cooled plates under 5 MPa load for 3 min. The obtained films were kept in a desiccator at room temperature for 24 h before characterization. For comparison, the pure PLLA-PEG-PLLA film was also prepared by the same conditions.

### 2.3. Characterization of PLLA-PEG-PLLA/ZnO Nanocomposites

Thermal transition properties of the samples were determined with a non-isothermal and isothermal differential scanning calorimeter (DSC) (PerkinElmer, Pyris Diamond, Waltham, MA, USA) under a nitrogen flow. For DSC heating scans, the samples were maintained at 200 °C for 3 min to remove thermal history before quickly quenching to 0 °C. The samples were then scanned from 0 °C to 200 °C at a heating rate of 10 °C/min under a nitrogen flow. The samples were then scanned from 200 °C to 0 °C at a cooling rate of 10 °C/min for DSC cooling scans.

The degree of crystallinity of the samples from DSC (*DSC-X_c_*) of PLLA crystallites was determined from DSC heating scans using the following equation.
*DSC-X_c_* (%) = [(Δ*H_m_* − Δ*H_cc_*)/(93.6 × *W_PLLA_*)] × 100(1)
where Δ*H_m_* and Δ*H_cc_* are the enthalpies of melting and cold crystallization, respectively, and 93.6 J/g is 100% *DSC-X_c_* of PLLA [34]. *W_PLLA_* is the weight fraction of PLLA.

For isothermal DSC scans, the samples were cooled from 200 °C to 120 °C at a rate of 50 °C/min after melting at 200 °C for 3 min to remove thermal history. The samples were then isothermally scanned at 120 °C until they were completely crystallized.

Thermal decomposition properties of the samples were characterized with a non-isothermal thermogravimetric analyzer (TGA) (TA-Instruments, SDT Q600, New Castle, DE, USA). For the TGA test, a sample (~10 mg) was scanned from 50 °C to 800 °C at a heating rate of 20 °C/min under a nitrogen flow.

Phase morphology of the film samples was observed from the cryo-fractured film surfaces with a scanning electron microscope (SEM) (JEOL, JSM-6460LV, Tokyo, Japan) at 20 kV. The film samples were sputter-coated with gold before SEM analysis.

Crystalline structures of the film samples were determined with a wide-angle X-ray diffractometer (XRD) (Bruker Corporation, D8 Advance, Karlsruhe, Germany) with a CuKα radiation of 40 kV and 40 mA. Scan speed was 3°/min. The degree of crystallinity of film samples from XRD (*XRD−X_c_*) was calculated using the following equation.
*XRD* − *X_c_* (%) = [(*A_c_*)/(*A_c_* + *A_a_*)] × 100(2)
where *A_c_* is the peak area of PLLA crystallites and *A_a_* is the halo area of the amorphous phase.

Mechanical properties comprising ultimate tensile stress, strain at break, and Young’s modulus of the film samples were measured with a universal testing machine (Dongguan Liyi Environmental Technology Co., Ltd., LY-1066B, Dongguan, China) according to the standard test method of American Society for Testing and Materials (ASTM) standard: designation of D882. Film samples with 10 mm width were tested at 25 °C and 50% relative humidity with a gauge length of 50 mm, a speed tensile test of 50 mm/min, and a load cell of 100 kg. The tensile properties were averaged from five measurements.

Opacity of the film samples was determined from absorbance at a wavelength of 600 nm (*A*_600_) measured with a UV–vis spectrophotometer (Agilent Technologies, Cary 60, Mulgrave, VIC, Australia). The following equation was used to calculate the opacity of film samples [35,36]. The opacity value was averaged from three determinations.
Opacity (mm^−1^) = *A*_600_/*X*(3)
where *A*_600_ is the film’s absorbance at a wavelength of 600 nm and *X* is the film’s thickness (mm).

The UV-barrier property of the film samples was determined with a UV–vis spectrophotometer (Agilent Technologies, Cary 60, Mulgrave, VIC, Australia) at a wavelength range of 200 nm to 550 nm [15].

Moisture uptake of the film samples (20 mm × 20 mm) was investigated by placing the film sample in a desiccator with 90 ± 5% relative humidity maintained with a saturated sodium chloride solution at 25 °C. At time intervals (1, 3, 6, 12, 24, and 48 h), the weight of film samples was determined. Prior to testing, the film samples were dried in a vacuum oven at room temperature for 48 h before weighing. The following equation was used to calculate the moisture uptake of film samples [37]. The averaged moisture uptake was obtained from three different measurements.
Moisture uptake (%) = [(*W_f_* − *W_i_*)/*W_i_*] × 100(4)
where *W_i_* and *W_f_* are the weights of film samples before and after test.

The antibacterial properties of the film samples were evaluated using Gram-negative *E. coli* TISTR 780 and Gram-positive *S. aureus* TISTR 1466. A single colony of each bacterium was propagated in nutrient broth (NB) under agitation at 37 °C for 18 h. Before utilization, the bacterial concentration was meticulously adjusted to an absorbance at 600 nm wavelength (*A*_600_) of 0.1. The film samples with 10 mm diameter were rendered sterile by submersion in 95% ethanol, followed by a drying period of 10 min under ultraviolet light. Subsequently, 100 µL of bacterial suspension, adjusted to *A*_600_ of 0.1, was evenly distributed onto nutrient agar (NA). The film samples were then placed on the surface of the NA, which had been spread with bacteria. The plates were incubated at 37 °C for 18 h after which the diameter of the inhibition zone for each film was meticulously measured (mm) and documented. The diameter of inhibition zone was averaged from triplicate tests.

## 3. Results and Discussion

### 3.1. Thermal Transition Properties

The thermal transition properties of the pure PLLA-PEG-PLLA and PLLA-PEG-PLLA/ZnO nanocomposites were investigated by isothermal and non-isothermal DSCs. The DSC heating curves of the samples are compared in Figure 2 and their DSC results are summarized in Table 1. It was found that *T_g_* and *T_m_* values were in the range of 29–32 °C and 157–160 °C, respectively. Figure 3 shows the *T_g_* regions of the DSC heating curves. The *T_cc_* and *DSC-X_c_* values of pure PLLA-PEG-PLLA were 81 °C and 13.4%, respectively. Table 1 shows that the *T_cc_* peaks shifted from 81 °C to 64 °C when the 1 wt% nano-ZnO was incorporated and the *T_cc_* peak disappeared when the 2 wt% nano-ZnO was incorporated. Shifting to lower temperatures of *T_cc_* peaks indicated that the cold crystallization of PLLA end-blocks can occur at lower temperatures [25,36]. The disappearance of *T_cc_* peaks in DSC heating scans of the nanocomposites suggested a heterogeneous nucleation effect had occurred by addition of a nucleating agent [34]. This result could indicate that PLLA crystallization was accelerated by the added nano-ZnO. In addition, the *DSC-X_c_* values of nanocomposites largely increased as the nano-ZnO content increased up to 2 wt%. This supports the suggestion that the nano-ZnO induced a heterogeneous nucleating effect for PLLA-PEG-PLLA, thereby increasing the number of heterogeneous nucleation sites [38].

However, the *T_cc_* peaks shifted to higher temperatures again and the *DSC-X_c_* values decreased again when the nano-ZnO content increased beyond 2 wt%. This may have been due to the ZnO nanoparticles becoming agglomerated to reduce the effective nucleation during PLLA-PEG-PLLA’s crystallization, which SEM analysis will subsequently verify. In addition, the excessive nano-ZnO in the PLLA crystalline phase may have caused the compact structure of PLLA crystals to break [36].

Figure 4 compares DSC cooling curves of the samples and their DSC results are summarized in Table 2. The pure PLLA-PEG-PLLA had *T_c_* and Δ*H_c_* values of 99 °C and 10.5 J/g, respectively. Table 2 also indicates that the *T_c_* peaks shifted to higher temperatures and the Δ*H_c_* values increased as the nano-ZnO content increased up to 2 wt%. This supports a nucleation effect for PLLA-PEG-PLLA occurring during DSC cooling scans, which was enhanced by the added nano-ZnO, as attested by the increasing *T_c_* and Δ*H_c_* values [38,39,40]. However, the *T_c_* peaks shifted to lower temperatures again and the Δ*H_c_* values decreased again when the nano-ZnO content exceeded 2 wt%. The results of DSC cooling curves confirmed that the nano-ZnO in the range of 1–2 wt% enhanced nucleating effects for PLLA-PEG-PLLA according to the results of the above DSC heating analysis.

The nucleation effect of added nano-ZnO on PLLA-PEG-PLLA matrices was also investigated from isothermal DSC scans at 120 °C as shown in Figure 5a. It can be seen that the isothermal crystallization-peaks significantly shifted to shorter times when the nano-ZnO content increased up to 2 wt%, suggesting that the crystallization of PLLA-PEG-PLLA was accelerated [41]. This was followed by the isothermal crystallization-peaks shifting to longer times again as the nano-ZnO content increased beyond 2 wt%, suggesting that the crystallization of PLLA-PEG-PLLA was decelerated. Polymer samples with a 50% relative crystallinity were obtained at the crystallization half-time (*t_1/2_*) through isothermal DSC scans as indicated in Figure 5b. The pure PLLA-PEG-PLLA had a *t_1/2_* value of 3.5 min. The resulting *t_1/2_* values of nanocomposites with nano-ZnO contents of 1 wt%, 2 wt%, 3 wt%, and 4 wt% were 2.0, 1.9, 2.4, and 2.9 min, respectively. It can be seen that the *t_1/2_* values steadily decreased when the nano-ZnO content increased up to 2 wt%, followed by increasing the *t_1/2_* value again as the nano-ZnO content increased beyond 2 wt%. The results of DSC isothermal scans confirmed that the nano-ZnO in the range of 1–2 wt% enhanced the heterogeneous nucleation effect for PLLA-PEG-PLLA according to the results of the above DSC heating and cooling analyses. From the DSC results, it can be concluded that the crystallization properties of the nanocomposites can be improved by low incorporation (up to 2 wt%) of nano-ZnO within the PLLA-PEG-PLLA matrix.

### 3.2. Thermal Decomposition Properties

The thermal decomposition properties of the pure PLLA-PEG-PLLA and PLLA-PEG-PLLA/ZnO nanocomposites were determined from the mean of TGA as shown in Figure 6. The TGA results are summarized in Table 3. Thermal decomposition behavior of the pure PLLA-PEG-PLLA exhibited two decomposition steps of PLLA (250–350 °C) and PEG (350–450 °C) blocks. The pure PLLA-PEG-PLLA had a decomposition temperature at 5% weight loss (5%-*T_d_*) from a thermogravimetric (TG) thermogram at 282 °C (Figure 6a). The 5%-*T_d_* values decreased as the nano-ZnO content increased. This suggests that the thermal stability of PLLA-PEG-PLLA matrices decreased as the nano-ZnO content increased. It has been reported that the nano-ZnO was not completely thermally decomposed at 800 °C [26]. Therefore, the residue weight or ash at 800 °C of the nanocomposite samples was due to nano-ZnOs. As would be expected, the residue weight at 800 °C of the nanocomposite samples steadily increased as the nano-ZnO content increased.

The thermal decomposition behavior of the samples was also determined from derivative TG (DTG) thermograms as shown in Figure 6b. Peaks of decomposition temperature at the maximum rate for PLLA blocks (*PLLA-T_d,max_*) and for PEG blocks (*PEG-T_d,max_*) were found. The resulting *PLLA-T_d,max_* and *PEG-T_d,max_* values are also summarized in Table 3, which shows that the *PLLA-T_d,max_* peaks were significantly shifted to lower temperatures as the nano-ZnO content increased. It may be hypothesized that the ZnO nanoparticles activated depolymerization of PLLA end-blocks [27,42,43], whereas the *PEG-T_d,max_* peaks slightly shifted to lower temperatures as the nano-ZnO content increased. The TGA results support the conclusion that the addition of nano-ZnOs reduced the thermal stability of the PLLA-PEG-PLLA. However, the TGA results indicated that these nanocomposites were stable up to 250 °C, thus melt processing at 190 °C should still be possible for use in packaging applications.

### 3.3. Phase Morphology

Figure 7 shows SEM images of cryo-fractured film surfaces which were used to investigate their phase morphology. In the figure, ZnO nanoparticles can be observed throughout the PLLA-PEG-PLLA matrices. The nanocomposite films containing 1 wt% and 2 wt% nano-ZnO in Figure 7a,b, respectively, exhibited a good distribution of the ZnO nanoparticles in PLLA-PEG-PLLA matrices, indicative of good phase compatibility between the ZnO nanoparticles and the PLLA-PEG-PLLA matrices. The SEM results supported a conclusion that a good distribution of 1 wt% and 2 wt% nano-ZnOs enhanced the crystallization properties of PLLA-PEG-PLLA, as described in the above DSC analysis.

However, the agglomeration of some ZnO nanoparticles was also observed when the nano-ZnO contents were 3 wt% and 4 wt%, as indicated by white circles in Figure 7d,e, respectively. The addition of high-content inorganic fillers usually results in the agglomeration of filler particles in polymer matrices due to differences in hydrophilicity between the filler and polymer matrices [26,27,44]. The nano-ZnO agglomerates then reduced the nucleation effective for PLLA-PEG-PLLA matrices, which supported the above DSC analysis.

### 3.4. Crystalline Structures

The XRD profiles in Figure 8 were used to investigate the crystalline structures of film samples. The XRD profile of pure PLLA-PEG-PLLA film in Figure 8a showed a small broad peak at the 2-theta of 16.9°, which was assigned to the PLLA crystals [40]. Intensities of this peak significantly increased as the nano-ZnO content increased up to 2 wt%, followed by a decrease in the peak intensities again as the nano-ZnO content increased beyond 2 wt%. For nanocomposite films containing 2 wt% and 3 wt% nano-ZnOs in Figure 8c,d, respectively, the characteristic peaks at the 2-theta of 14.9°, 19.1°, and 22.5° due to the PLLA crystals were also detected [41,45]. Therefore, the crystalline structures of the PLLA end-blocks in PLLA-PEG-PLLA did not change upon the addition of nano-ZnOs.

The pure PLLA-PEG-PLLA film had an *XRD-X_c_* value of 10.3%. The *XRD-X_c_* values of PLLA-PEG-PLLA film matrices increased to 19.4% and 39.0% as 1 wt% and 2 wt% nano-ZnOs were incorporated, respectively. This supports the contention that nano-ZnOs enhanced the nucleation effect on film matrices according to the above DSC analysis. However, the *XRD-X_c_* values decreased again to 27.6% and 14.4% as the nano-ZnO contents were 3 wt% and 4 wt%, respectively. This may be explained by the nano-ZnOs having agglomerated [27] as described in the above SEM analysis. The XRD results confirmed that nano-ZnOs can act as an effective nucleating agent for PLLA-PEG-PLLA when the nano-ZnO content does not exceed 2 wt%.

### 3.5. Tensile Properties

Tensile properties of film samples (stress at yield, stress at break, strain at break, and Young’s modulus) were determined from stress–strain curves as shown in Figure 9. The tensile results are summarized in Table 4. The tensile properties of the nanocomposite film containing 4 wt% nano-ZnOs could not be measured because of its brittleness. From stress–strain curves, it is seen that all film samples exhibited a yield point, suggesting that all the film samples were highly flexible. From Table 4, the pure PLLA-PEG-PLLA film is seen to have had a stress at a yield of 21.2 MPa, a stress at break of 17.4 MPa, a strain at break of 102%, and a Young’s modulus of 312 MPa. When the 1 wt% and 2 wt% nano-ZnOs were incorporated, the values of stress at yield, stress at break, and Young’s modulus of nanocomposite films significantly increased compared to the pure PLLA-PEG-PLLA film. This indicates the added ZnO nanoparticles induced a reinforcing effect that improved these tensile properties of PLLA-PEG-PLLA matrices. The fine distribution of filler particles in the polymer matrix and good interfacial adhesion between the filler particles and the polymer matrix enhanced the mechanical properties of the polymer matrices [13,36,44].

However, the values of stress at yield, stress at break, and Young’s modulus of nanocomposite films dramatically decreased when 3 wt% nano-ZnOs were incorporated compared to the pure PLLA-PEG-PLLA film. This may be explained by the agglomeration of nano-ZnOs at a high content, leading to the non-homogeneity of the nanocomposite films that reduced stress transfer between the polymer matrices and filler [46,47]. The strains at break of PLLA-PEG-PLLA films steadily decreased as the ZnO content increased compared to the pure PLLA-PEG-PLLA film because the reinforcing effect of inorganic filler limits the mobility and ductile flow of the polymer chains [44,48]. It should be noted that these nanocomposite films still had more flexibility than PLLA films because they still had a higher strain at break (29–57%) than the pure PLLA film (about 5%) [33].

### 3.6. Opacity and UV-Barrier Property

The opacity of film samples was calculated from Equation (3), as also summarized in Table 4. The pure PLLA-PEG-PLLA film had an opacity of 0.29 mm^−1^. The film’s opacity dramatically increased up to 1.19 mm^−1^ when the 1 wt% nano-ZnO was incorporated. The addition of nano-ZnO filler increased the opacity of polymer-based films because of the opaque character of nano-ZnOs, which has been reported in earlier works [15,25]. The opacities of films increased as the ZnO content increased. As shown in Figure 10, PLLA-PEG-PLLA/ZnO nanocomposite films appeared to have higher opacity than the pure PLLA-PEG-PLLA film. However, the words obscured by the composite films were still readable and easily visible. Thus, these nanocomposite films could be used as food packaging where the food characteristics are still visible.

The absorbance in the 200–550 nm wavelength range was used to determine the UV-barrier properties of pure PLLA-PEG-PLLA and the nanocomposite films, as illustrated in Figure 11. It was found that the absorbance of film samples in the 200–550 nm wavelength range increased as the ZnO content increased, indicating that the UV-barrier properties of the PLLA-PEG-PLLA films were improved by the addition of nano-ZnOs. This is because the nano-ZnO is a UV absorber [15,49,50,51]. The addition of nano-ZnOs can improve the effectiveness of the PLLA-PEG-PLLA matrix to shield it from UV radiation. Therefore, the resulting nanocomposites could be used as UV-shielding food packaging.

### 3.7. Moisture Uptake

Figure 12 shows the moisture uptake of the pure PLLA-PEG-PLLA and nanocomposite films for 48 h. It can be seen that the equilibrium values were reached after an incubation time of 24 h. The pure PLLA-PEG-PLLA film exhibited the highest moisture uptake. The addition of nano-ZnOs decreased the moisture uptake of the film. Moisture uptake decreased as the nano-ZnO content increased. This may have been due to the added nano-ZnOs extending the tortuous diffusion pathways of the water vapor, thereby enhancing the performance of the water vapor barrier in the PLLA-PEG-PLLA matrix [1,36]. In addition, the increasing crystallinity of the PLLA-PEG-PLLA matrix caused by the addition of nano-ZnOs may also have led to the extensive tortuosity of the water vapor transport pathways [23].

### 3.8. Antibacterial Properties

The antibacterial properties of the film samples were evaluated by quantifying the growth inhibition of *E. coli* (Gram-negative bacteria) and *S. aureus* (Gram-positive bacteria) as shown in Figure 13. These bacteria were selected as representative bacteria in this work because they have been recognized as hospital-associated infections [52]. The antibacterial test resulted in a clear circular area around the film samples which was an inhibition zone in which the bacterial colonies did not grow. The diameter of the inhibition zone directly related to the antibacterial ability of the film samples. Inhibition zones were clearly observed for all the nanocomposite films but not for the pure PLLA-PEG-PLLA film. This suggests that the pure PLLA-PEG-PLLA film had no antibacterial ability, whereas all the nanocomposite films can inhibit and prevent the growth of both types of bacteria. Therefore, the addition of nano-ZnOs introduced antibacterial abilities to the PLLA-PEG-PLLA films.

The diameters of the inhibition zones of film samples are summarized in Table 5. The inhibition zone diameter of the nanocomposite films increased as the nano-ZnO content increased. The results indicated that the antibacterial ability of the films increased with the nano-ZnO content. This was due to the high antibacterial ability of nano-ZnOs [1,13,51]. This may involve the integrity of the bacteria cells being damaged by the release of Zn^2+^ ions and by photocatalytic reactions yielding reactive oxygen species (ROS) [15,26,53]. In Table 5, it is seen that the diameters of inhibition zones for *S. aureus* bacteria of the nanocomposite films were higher than *E. coli* bacteria for the same nano-ZnO content. This suggests the nanocomposite films showed greater antibacterial ability against Gram-positive *S. aureus* than the Gram-negative *E. coli*, which concurs with the previous report in the literature [15,53]. This is due to differences in the cell-wall structures of these bacteria [53].

## 4. Conclusions

In this study, the effects of adding nano-ZnOs to flexible PLLA-PEG-PLLA were investigated in terms of crystallization properties, thermal stability, phase compatibility, tensile properties, film opacity, UV-barrier properties, and antibacterial properties. The crystallization properties of PLLA-PEG-PLLA matrices were enhanced by the addition of nano-ZnOs as follows. The *T_cc_* peaks of the PLLA-PEG-PLLA matrices shifted to a lower temperature and disappeared, the *T_c_* peaks shifted to a higher temperature, the *t_1/2_* values decreased, and both the *DSC-X_c_* and the *XRD-X_c_* values increased as the nano-ZnO content increased, suggesting that the ZnO nanoparticles induced a heterogeneous nucleation effect. However, the crystallization properties of PLLA-PEG-PLLA matrices decreased again as the nano-ZnO content increased beyond 2 wt% due to the agglomeration of some ZnO nanoparticles.

The thermal stability of PLLA-PEG-PLLA matrices decreased as the nano-ZnO content increased, as indicated from the shifting of the *PLLA-T_d,max_* peaks to lower temperatures. The tensile properties of PLLA-PEG-PLLA films were improved by the addition of nano-ZnOs as follows. The stress at yield, stress at break, and Young’s modulus of PLLA-PEG-PLLA films increased as the 1 wt% and 2 wt% nano-ZnOs were incorporated, suggesting that the nano-ZnOs induced a reinforcing effect. However, these tensile properties decreased again as the nano-ZnO content was increased beyond 2 wt% due to some ZnO nanoparticles becoming agglomerated. The opacity, UV-barrier property, and antibacterial ability of the nanocomposite films increased and the moisture uptake decreased as the nano-ZnO content increased. All the nanocomposite films showed clearly quantifiable growth inhibition of *E. coli* (Gram-negative bacteria) and *S. aureus* (Gram-positive bacteria) but the pure PLLA-PEG-PLLA film had no antibacterial ability. It can be concluded that the nano-ZnO shows promise as a multi-functional filler to improve the crystallization, mechanical, UV-barrier, and antibacterial ability of the flexible PLLA-PEG-PLLA. The PLLA-PEG-PLLA/ZnO nanocomposites have the potential for use as novel flexible, biodegradable, and antibacterial food packaging. Nonetheless, before these nanocomposites can be industrialized, further investigation is required to fully understand the nano-ZnO migration, biodegradation, cytotoxicity, and permeation properties of O_2_, CO_2_, and water vapor.

## Figures and Tables

**Figure 1 polymers-16-01660-f001:**
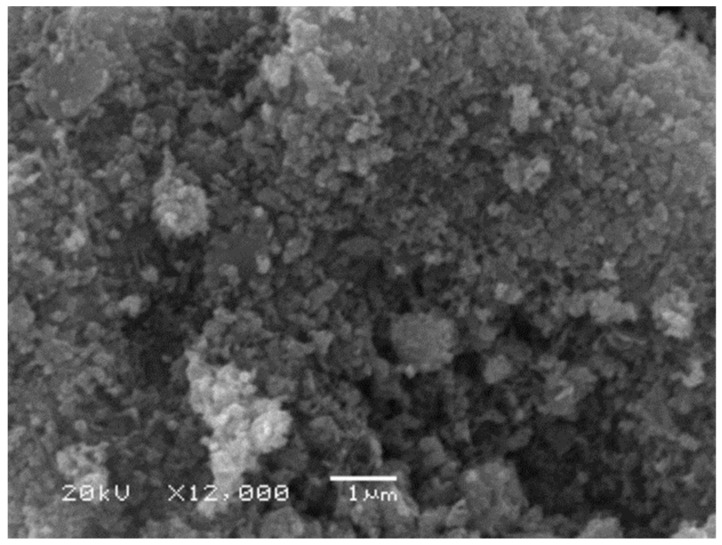
SEM image of ZnO nanoparticles (bar scale = 1 µm).

**Figure 2 polymers-16-01660-f002:**
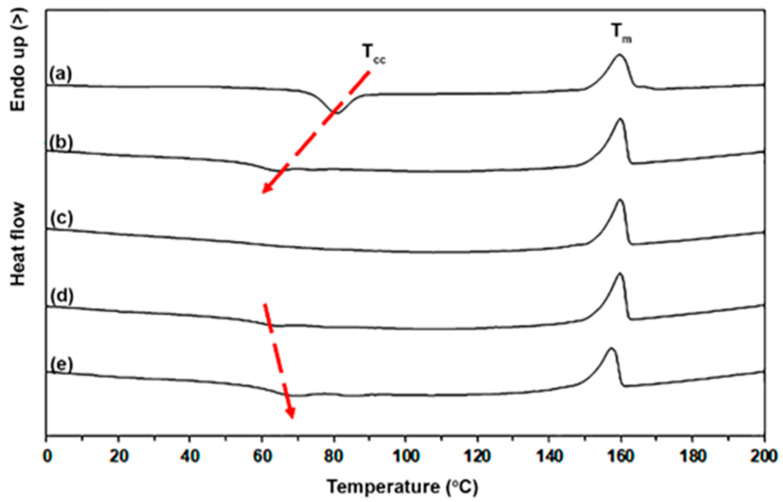
DSC heating curves of (a) pure PLLA-PEG-PLLA and PLLA-PEG-PLLA/ZnO nanocomposites with nano-ZnO contents of (b) 1 wt%, (c) 2 wt%, (d) 3 wt%, and (e) 4 wt% (red arrows show shifting of *T_cc_* peaks).

**Figure 3 polymers-16-01660-f003:**
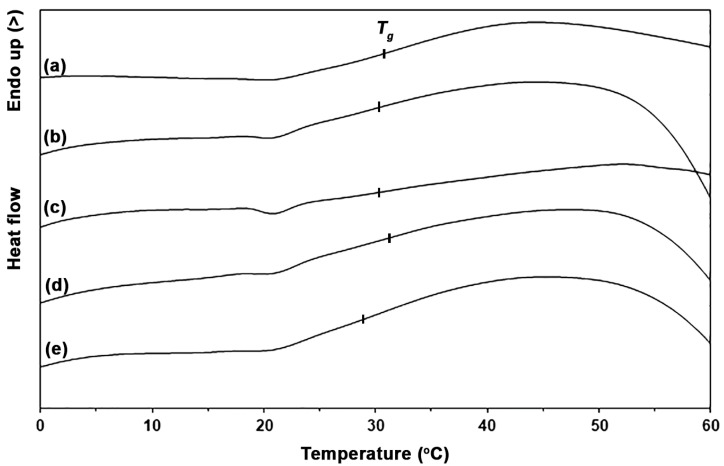
*T_g_* regions of DSC heating curves of (a) pure PLLA-PEG-PLLA and PLLA-PEG-PLLA/ZnO nanocomposites with nano-ZnO contents of (b) 1 wt%, (c) 2 wt%, (d) 3 wt%, and (e) 4 wt%.

**Figure 4 polymers-16-01660-f004:**
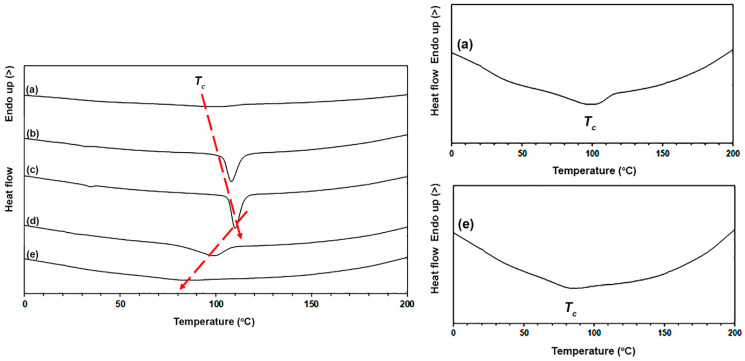
DSC cooling curves of (a) pure PLLA-PEG-PLLA and PLLA-PEG-PLLA/ZnO nanocomposites with nano-ZnO contents of (b) 1 wt%, (c) 2 wt%, (d) 3 wt%, and (e) 4 wt% (red arrows show shifting of *T_c_* peaks; Expanded curves of (a) and (e) as shown).

**Figure 5 polymers-16-01660-f005:**
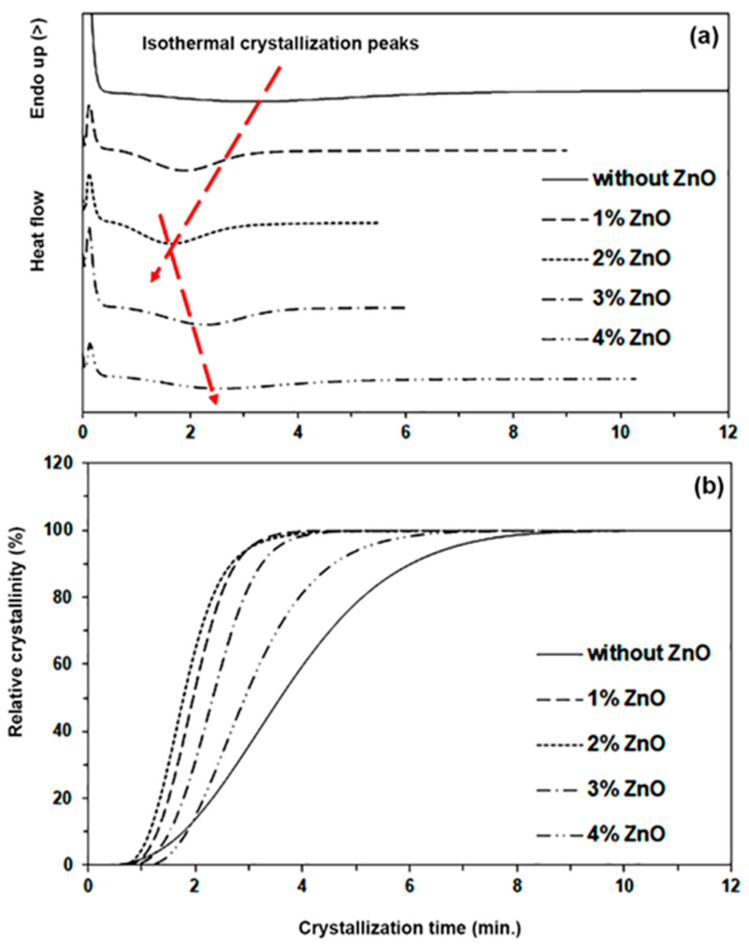
(**a**) Isothermal crystallization curves at 120 °C and (**b**) relative crystallinity–crystallization time curves of pure PLLA-PEG-PLLA and PLLA-PEG-PLLA/ZnO nanocomposites with various contents of nano-ZnOs (red arrows show shifting of isothermal crystallization-peaks).

**Figure 6 polymers-16-01660-f006:**
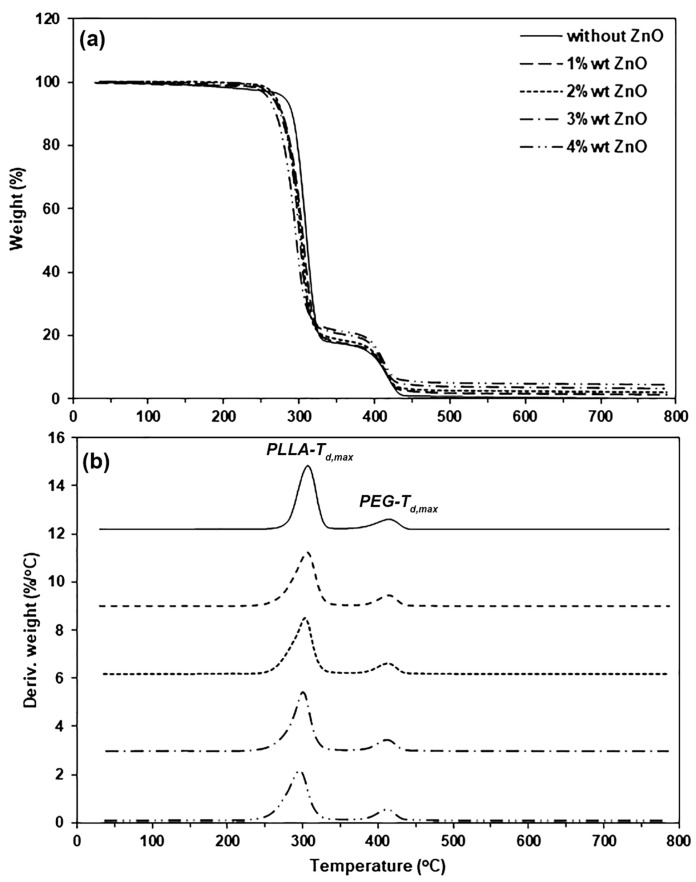
(**a**) TG and (**b**) DTG thermograms of pure PLLA-PEG-PLLA and PLLA-PEG-PLLA/ZnO nanocomposites with various contents of nano-ZnOs.

**Figure 7 polymers-16-01660-f007:**
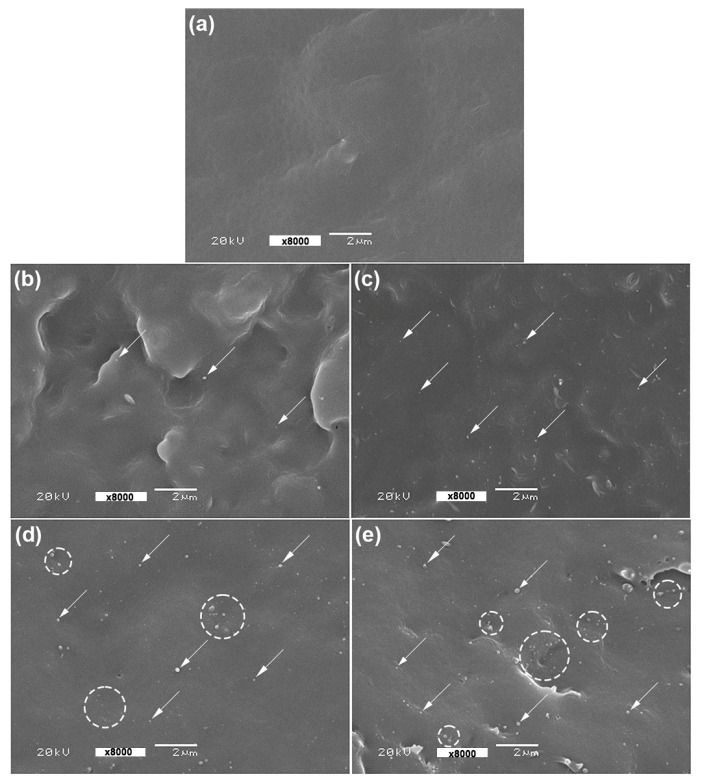
SEM images of cryo-fractured surfaces of (**a**) pure PLLA-PEG-PLLA film and PLLA-PEG-PLLA/ZnO nanocomposite films with nano-ZnO contents of (**b**) 1 wt%, (**c**) 2 wt%, (**d**) 3 wt%, and (**e**) 4 wt% (some ZnO nanoparticles were indicated by white arrows and some nano-ZnO agglomerates were indicated by white circles, all bar scales = 2 µm).

**Figure 8 polymers-16-01660-f008:**
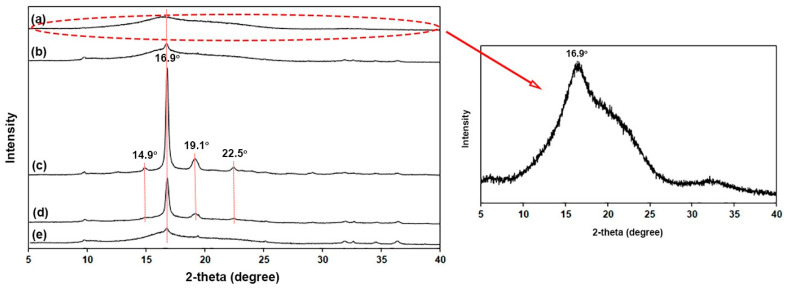
XRD profiles of (a) pure PLLA-PEG-PLLA film and PLLA-PEG-PLLA/ZnO nanocomposite films with nano-ZnO contents of (b) 1 wt%, (c) 2 wt%, (d) 3 wt%, and (e) 4 wt%.

**Figure 9 polymers-16-01660-f009:**
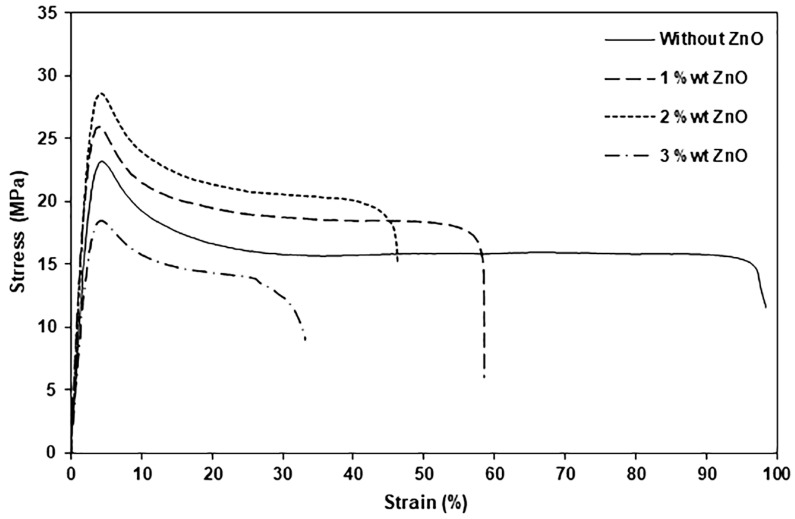
Selected stress–strain curves of pure PLLA-PEG-PLLA and PLLA-PEG-PLLA/ZnO nanocomposite films with various contents of ZnOs.

**Figure 10 polymers-16-01660-f010:**
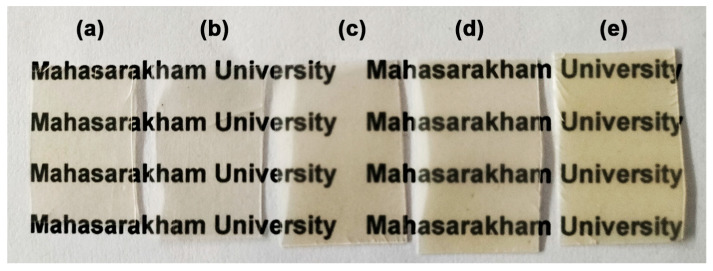
Photograph of (a) pure PLLA-PEG-PLLA film and PLLA-PEG-PLLA/ZnO nanocomposite films with nano-ZnO contents of (b) 1 wt%, (c) 2 wt%, (d) 3 wt%, and (e) 4 wt%.

**Figure 11 polymers-16-01660-f011:**
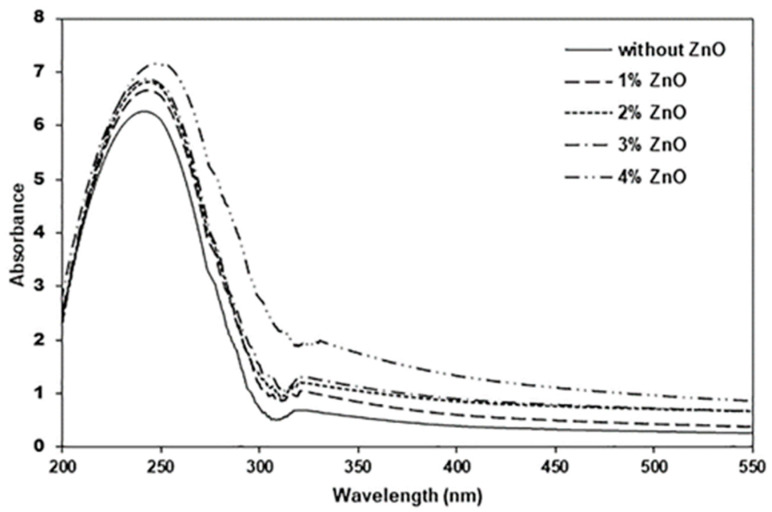
UV–vis spectra in UV region of pure PLLA-PEG-PLLA film and PLLA-PEG-PLLA/ZnO nanocomposite films with various contents of nano-ZnOs.

**Figure 12 polymers-16-01660-f012:**
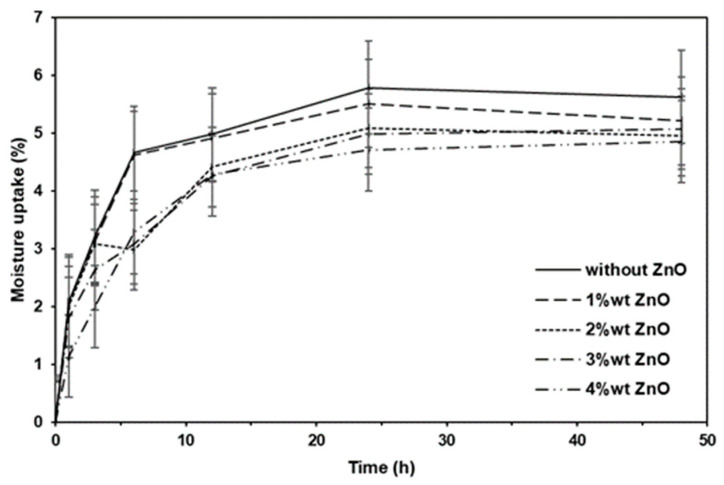
Moisture uptakes of pure PLLA-PEG-PLLA film and PLLA-PEG-PLLA/ZnO nanocomposite films with various contents of nano-ZnOs.

**Figure 13 polymers-16-01660-f013:**
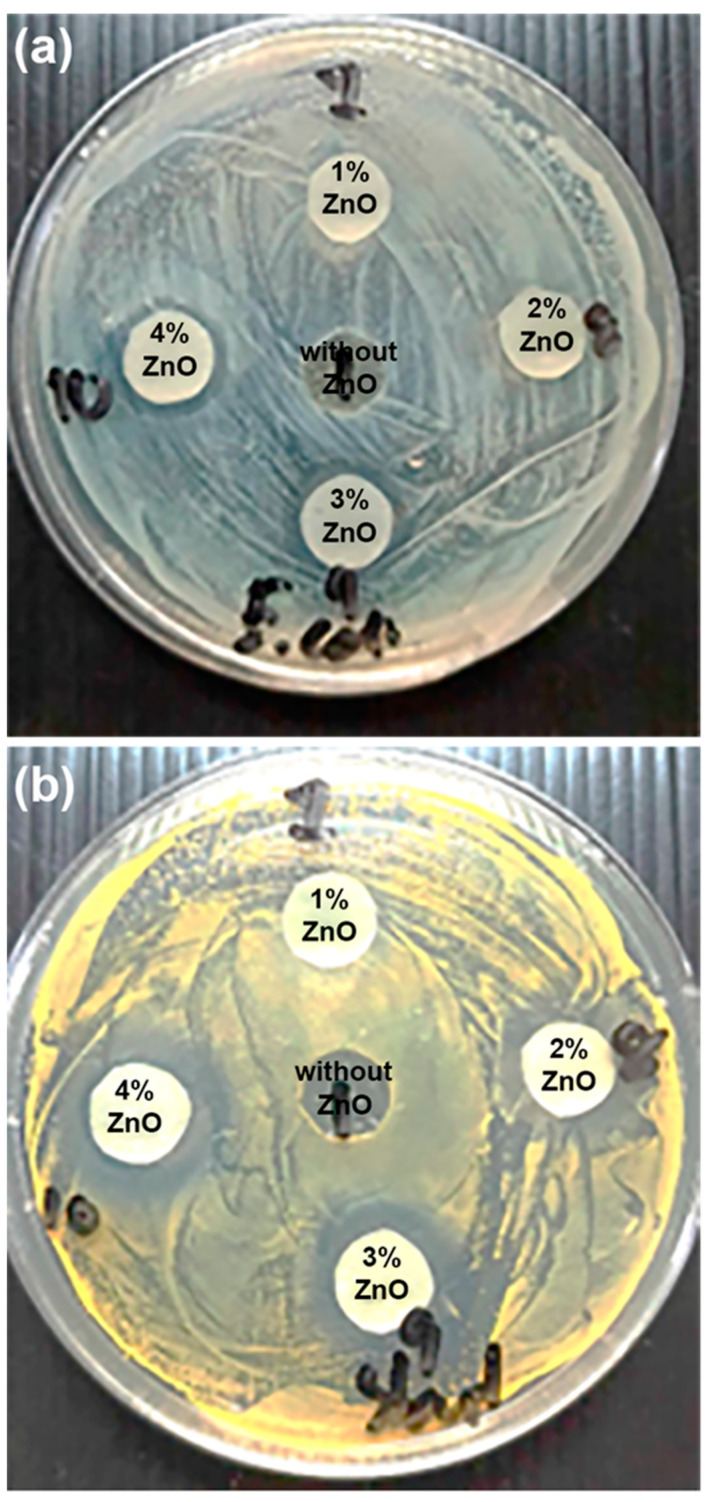
Agar disc diffusion of antibacterial activity against (**a**) Gram-negative *E. coli* TISTR 780 and (**b**) Gram-positive *S. aureus* TISTR 1466 of pure PLLA-PEG-PLLA film without nano-ZnOs and PLLA-PEG-PLLA/ZnO nanocomposite films with various contents of nano-ZnOs.

**Table 1 polymers-16-01660-t001:** Results from DSC heating curves of pure PLLA-PEG-PLLA and PLLA-PEG-PLLA/ZnO nanocomposites.

Nano-ZnO Content (wt%)	*T_g_* (°C)	*T_cc_* (°C)	Δ*H_cc_* (J/g)	*T_m_* (°C)	Δ*H_m_* (J/g)	*DSC-X_c_* (%)
-	32	81	20.9	159	31.3	13.4
1	31	64	9.6	160	31.1	27.9
2	31	-	-	160	30.8	40.4
3	32	64	8.4	160	30.9	29.9
4	29	68	16.3	157	31.2	20.0

**Table 2 polymers-16-01660-t002:** The results from DSC cooling curves of pure PLLA-PEG-PLLA and PLLA-PEG-PLLA/ZnO nanocomposites.

Nano-ZnO Content (wt%)	*T_c_* (°C)	Δ*H_c_* (J/g)
-	99	10.5
1	108	31.2
2	110	32.2
3	99	29.5
4	83	8.5

**Table 3 polymers-16-01660-t003:** Thermal decomposition properties of pure PLLA-PEG-PLLA and PLLA-PEG-PLLA/ZnO nanocomposites.

Nano-ZnO Content (wt%)	5%-*T_d_* ^1^ (°C)	Residue Weight at 800 °C ^1^ (%)	*PLLA-T_d,max_* ^2^ (°C)	*PEG-T_d,max_* ^2^ (°C)
-	282	0.4	310	416
1	269	1.3	308	415
2	266	2.2	304	415
3	264	3.5	300	413
4	261	4.5	296	413

^1^ Obtained from TG thermograms. ^2^ Obtained from DTG thermograms.

**Table 4 polymers-16-01660-t004:** Tensile properties and opacity of the pure PLLA-PEG-PLLA and PLLA-PEG-PLLA/ZnO nanocomposite films.

Nano-ZnO Content (wt%)	Stress at Yield (MPa)	Stress at Break (MPa)	Strain at Break (%)	Young’s Modulus (MPa)	Opacity (mm^−1^)
-	21.2 ± 2.5	17.4 ± 3.3	102 ± 8	312 ± 27	0.29 ± 0.02
1	27.6 ± 3.1	18.2 ± 4.8	57 ± 6	396 ± 25	1.19 ± 0.05
2	29.4 ± 2.2	21.3 ± 4.1	41 ± 4	406 ± 34	1.56 ± 0.04
3	17.8 ± 3.6	14.7 ± 4.7	29 ± 6	274 ± 18	1.64 ± 0.09
4	- *	- *	- *	- *	2.10 ± 0.12

* PLLA-PEG-PLLA/4 wt% ZnO nanocomposite film could not be cut for tensile test because of its brittleness.

**Table 5 polymers-16-01660-t005:** Antibacterial properties of pure PLLA-PEG-PLLA and PLLA-PEG-PLLA/ZnO nanocomposite films.

Nano-ZnO Content (wt%)	Inhibition Zone (mm)	
	*E. coli* TISTR 780	*S. aureus* TISTR 1466
0	0	0
1	12.3 ± 0.6	12.7 ± 0.6
2	15.3 ± 0.6	18.7 ± 0.6
3	15.7 ± 0.7	19.3 ± 0.5
4	16.0 ± 0.1	22.3 ± 0.6

## Data Availability

Data are contained within the article.

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
