# Peer review of "Preparation of Poly(L-lactide)-b-poly(ethylene glycol)-b-poly(L-lactide)/Zinc Oxide Nanocomposite Bioplastics for Potential Use as Flexible and Antibacterial Food Packaging"

_polymers, 2024, doi:10.3390/polym16121660_

Round 1
Reviewer 1 Report
Comments and Suggestions for Authors
The article is devoted to the development of PLLA-PEG-PLLA-based composites with ZnO2 addition for packaging applications. From the advantages of the article we can single out that the independently synthesized copolymer PLLA-PEG-PLLA was used. The addition of PEG avoids the brittleness of PLLA. Also, the effects of ZnO2 oxide addition on the supramolecular structure of the composite are discussed in the paper. It is shown by different methods that at addition of more than 2% the particles start to aggerize. This affects the properties of the obtained composites. There are several questions to the authors of the publication:
1) The authors position their samples as films for food packaging. The samples in the article are obtained by pressing, while packaging films are usually obtained by blow or cast extrusion. Other effects are possible with these methods.
2) When positioning the samples as packaging samples, it is necessary to investigate their permeability to O2, CO2, and water vapor. These parameters can be used to evaluate the ability of the films to preserve the freshness of the products.
3) It is also necessary to carry out a migration test. This test shows the migration of substances from the sample to the packaged product. Since in this work ZnO2 is used, which can be toxic in certain concentrations, its migration and poisoning of the product is probable.
4) The authors do not address the disposal and biodegradation of the samples obtained. Probably, due to the addition of PEG, the rate of hydrolysis will increase. It is interesting to know the authors' opinion on this matter.
In general, the article is very interesting, but I advise the authors to change the positioning of their samples from packaging films to biodegradable composites with antibacterial properties. After the change of focus, the article certainly deserves to be published in the journal Polymers. And in subsequent articles, after conducting tests showing the properties of the films as packaging materials, the samples can be proposed for packaging.
Author Response
Reviewer # 1_Round 1
Manuscript ID: polymers-3046225
Title: Preparation of poly(L-lactide)-b-poly(ethylene glycol)-b-poly(L-lactide)/zinc oxide nanocomposite bioplastics for potential use as an flexible and antibacterial food packaging
Authors: Yaowalak Srisuwan, Prasong Srihanam, Surachai Rattanasuk, Yodthong Baimark *
Reviewer # 1
The article is devoted to the development of PLLA-PEG-PLLA-based composites with ZnO2 addition for packaging applications. From the advantages of the article we can single out that the independently synthesized copolymer PLLA-PEG-PLLA was used. The addition of PEG avoids the brittleness of PLLA. Also, the effects of ZnO2 oxide addition on the supramolecular structure of the composite are discussed in the paper. It is shown by different methods that at addition of more than 2% the particles start to aggerize. This affects the properties of the obtained composites. There are several questions to the authors of the publication:
In general, the article is very interesting, but I advise the authors to change the positioning of their samples from packaging films to biodegradable composites with antibacterial properties. After the change of focus, the article certainly deserves to be published in the journal Polymers. And in subsequent articles, after conducting tests showing the properties of the films as packaging materials, the samples can be proposed for packaging.
Authors: The authors would like to sincerely thank the reviewer for the time that the reviewer spent reading the paper and for their perceptive comments. All the comments have been used to improve the paper. A detailed point-by-point set of responses to the reviewer inputs is provided. All corrections are highlighted in red
Question 1. The authors position their samples as films for food packaging. The samples in the article are obtained by pressing, while packaging films are usually obtained by blow or cast extrusion. Other effects are possible with these methods.
Authors: Yes, blow and cast extrusion are usually the methods used for producing packaging products. In this work, we provide an introduction to research on effects of nano-ZnO addition on various properties of PLLA-PEG-PLLA. However, the compressed film samples can be used to determine the properties of nanocomposites.
Question 2. When positioning the samples as packaging samples, it is necessary to investigate their permeability to O2, CO2, and water vapor. These parameters can be used to evaluate the ability of the films to preserve the freshness of the products.
Authors: We agree that the permeability of O2, CO2, and water vapor affects the freshness of food products. However, we apologize for the analysis of the permeability properties of the film samples. Due to the unavailability of our instruments at this time, we have also noted this issue on P. 16, lines 507-510 of the revised manuscript as follows.
Nonetheless, before these nanocomposites can be industrialized, further investigation is required to fully understand the nano-ZnO migration, biodegradation, cytotoxicity, and permeation properties of O2, CO2, and water vapor.
Question 3. It is also necessary to carry out a migration test. This test shows the migration of substances from the sample to the packaged product. Since in this work ZnO2 is used, which can be toxic in certain concentrations, its migration and poisoning of the product is probable.
Authors: Yes, the long-term impact of nano-ZnO migration on human health and the environment is concerning and remains unresolved. It is necessary to test it before use in food packaging. The migration test of nano-ZnO from the film samples is not available in this work. However, we have noted this issue on P. 16, lines 507-510 of the revised manuscript as follows.
Nonetheless, before these nanocomposites can be industrialized, further investigation is required to fully understand the nano-ZnO migration, biodegradation, cytotoxicity, and permeation properties of O2, CO2, and water vapor.
Question 4. The authors do not address the disposal and biodegradation of the samples obtained. Probably, due to the addition of PEG, the rate of hydrolysis will increase. It is interesting to know the authors' opinion on this matter.
Authors: Yes, the hydrophilic PEG middle-blocks accelerated the biodegradation in soil of PLLA-PEG-PLLA, as described in our previous work [reference no. 31]. The pure PLLA-PEG-PLLA exhibited faster biodegradation in soil than the pure PLLA. However, in this work, the biodegradation in soil of the nanocomposites has not been determined. However, we have noted this issue on P. 16, lines 507-510 of the revised manuscript as follows.
Nonetheless, before these nanocomposites can be industrialized, further investigation is required to fully understand the nano-ZnO migration, biodegradation, cytotoxicity, and permeation properties of O2, CO2, and water vapor.
Concluding Remarks
We hope that our responses answer the reviewers’ comments to their satisfaction and that the revisions that have been made to the paper enhance its clarity for the benefit of the reader.
Yours Faithfully,
The Authors

Reviewer 2 Report
Comments and Suggestions for Authors
Review Report _ polymers-3046225-peer-review-v1
I have gone through the manuscript entitled “Preparation of Poly(L-lactide)-b-Poly (ethylene glycol)-b-Poly (L-2 lactide)/Zinc Oxide Nanocomposite Bioplastics for Potential Use as a Flexible and Antibacterial Food Packaging’’. The authors need to address the following points before acceptance of the manuscript.
1. The authors have mentioned that the opacity of the film samples was determined from the absorbance at a wavelength of 600 nm (A600) and the absorbance in the 200−550 nm wavelength range was used to determine the UV barrier properties, need to explain.
2. The Tg values from the Figs. 2 and 3 are not clear. So, it is suggested to plot the graph within this Tg range separately.
3. Similarly, in Fig. 3, the Tc peak values for plots (a) and (e) are not clear.
4. In thermal decomposition properties section, the authors have mentioned that “This suggests that thermal stability of PLLA-PEG-PLLA matrices was reduced as the nano-ZnO content increased. The residue weight at 800 °C of the samples from TG thermograms steadily increased as the nano-ZnO content increased because the nano-ZnO had high thermal stability and was not completely thermally decomposed at 800 °C” – this statement is not clear. Authors need to explain this properly by also considering the effect of increasing weight percent of nano-ZnO.
5. In XRD analysis, kindly mention the 2θ peak values also within the figure.
6. For pure PLLA-PEG-PLLA, the stress at break and strain at break values are not matching with the figure’s stress-strain curve. Kindly correct it.
7. You have shown that the young’s modulus value of 2 wt% loaded ZnO nanocomposite has increased compared to 1 wt% loaded ZnO nanocomposite in Table 4 but in Figure 8, it is showing that the young’s modulus value of 2 wt% loaded ZnO nanocomposite has decreased compared to 1 wt% loaded ZnO nanocomposites. Kindly correct it.
8. As you have mentioned in Table 4 that the opacity value of 4 wt% loaded ZnO nanocomposite is higher but in Figure 9, it is showing that 3 wt% loaded ZnO nanocomposite is more opaque than 4 wt% loaded ZnO nanocomposite. Kindly correct it.
9. In conclusion section, authors need to mention the novelty of their work.
10. Authors need to check the English language throughout the manuscript and need to correct it.

Minor correction in English language is required.
Author Response
Reviewer # 2_Round 1
Manuscript ID: polymers-3046225
Title: Preparation of poly(L-lactide)-b-poly(ethylene glycol)-b-poly(L-lactide)/zinc oxide nanocomposite bioplastics for potential use as an flexible and antibacterial food packaging
Authors: Yaowalak Srisuwan, Prasong Srihanam, Surachai Rattanasuk, Yodthong Baimark *
Reviewer # 2
I have gone through the manuscript entitled “Preparation of Poly(L-lactide)-b-Poly (ethylene glycol)-b-Poly (L-2 lactide)/Zinc Oxide Nanocomposite Bioplastics for Potential Use as a Flexible and Antibacterial Food Packaging’’. The authors need to address the following points before acceptance of the manuscript.
Authors: The authors would like to sincerely thank the reviewer for the time that the reviewer spent reading the paper and for their perceptive comments. All the comments have been used to improve the paper. A detailed point-by-point set of responses to the reviewer inputs is provided. All corrections are highlighted in red.
Question 1. The authors have mentioned that the opacity of the film samples was determined from the absorbance at a wavelength of 600 nm (A600) and the absorbance in the 200−550 nm wavelength range was used to determine the UV barrier properties, need to explain.
Authors: The absorbance at 600 nm wavelength (A600) was used to determine opacity of film samples corresponding to reference nos. 35 and 36, as described on P. 4, lines 160-168 of the revised manuscript, whereas, the absorbance in range 200-550 nm wavelength (UVA and UVB region wavelength) was used to determined the UV barrier properties corresponding to reference no. 15, as described on P. 4, lines 170-172 of the revised manuscript as follows.
Opacity of the film samples was determined from absorbance at a wavelength of 600 nm (A600) measured with a UV-vis spectrophotometer (Agilent Technologies, Cary 60, Victoria, Australia). The following equation was used to calculate the opacity of film samples [35,36]. The opacity value was averaged from three determinations.
Opacity (mm-1) = A600/X (3)
where A600 is the film’s absorbance at a wavelength of 600 nm and X is the film’s thickness (mm).
The UV-barrier property of the film samples was determined with a UV-vis spectrophotometer (Agilent Technologies, Cary 60, Victoria, Australia) at a wavelength range of 200 nm to 550 nm [15].
Question 2. The Tg values from the Figs. 2 and 3 are not clear. So, it is suggested to plot the graph within this Tg range separately.
Authors: The Tg region of DSC curves have been expanded as shown in Figure 3 on P. 6 of the revised manuscript as follows.
Figure 3. Tg regions of DSC heating curves of (a) pure PLLA-PEG-PLLA and PLLA-PEG-PLLA/ZnO nanocomposites with nano-ZnO contents of (b) 1 %wt, (c) 2 %wt, (d) 3 %wt, and (e) 4 %wt.
Question 3. Similarly, in Fig. 3, the Tc peak values for plots (a) and (e) are not clear.
Authors: The cooling curves of curves (a) and (e) have been expanded for observing the Tc peaks as shown in Figure 4 on P. 7 of the revised manuscript as follows.
Figure 4. DSC cooling curves of (a) pure PLLA-PEG-PLLA and PLLA-PEG-PLLA/ZnO nanocomposites with nano-ZnO contents of (b) 1 %wt, (c) 2 %wt, (d) 3 %wt, and (e) 4 %wt (red arrows show shifting of Tc peaks; Expanded curves of (a) and (e) as shown).
Question 4. In thermal decomposition properties section, the authors have mentioned that “This suggests that thermal stability of PLLA-PEG-PLLA matrices was reduced as the nano-ZnO content increased. The residue weight at 800 °C of the samples from TG thermograms steadily increased as the nano-ZnO content increased because the nano-ZnO had high thermal stability and was not completely thermally decomposed at 800 °C” – this statement is not clear. Authors need to explain this properly by also considering the effect of increasing weight percent of nano-ZnO.
Authors: This statement has re-written on P. 8, lines 285-290 of the revised manuscript as follows.
This suggests that thermal stability of PLLA-PEG-PLLA matrices decreased as the nano-ZnO content increased. It has been reported that the nano-ZnO was not completely thermally decomposed at 800 °C [26]. Therefore, the residue weight or ash at 800 °C of the nanocomposite samples was due to nano-ZnO. As would be expected, the residue weight at 800 °C of the nanocomposite samples steadily increased as the nano-ZnO content increased.
Question 5. In XRD analysis, kindly mention the 2θ peak values also within the figure.
Authors: The 2θ peak values have been assigned in Figure 8 on P. 11 of the revised manuscript as follows.
Figure 8. XRD profiles of (a) pure PLLA-PEG-PLLA film and PLLA-PEG-PLLA/ZnO nanocomposite films with nano-ZnO contents of (b) 1 %wt, (c) 2 %wt, (d) 3 %wt, and (e) 4 %wt.
Question 6. For pure PLLA-PEG-PLLA, the stress at break and strain at break values are not matching with the figure’s stress-strain curve. Kindly correct it.
Authors: The stress at break and strain at break values from selected tensile curve of pure PLLA-PEG-PLLA in Figure 9 were in range of average values (± S.D.) of stress at break and strain at break in Table 4 of the revised manuscript.
Question 7. You have shown that the young’s modulus value of 2 wt% loaded ZnO nanocomposite has increased compared to 1 wt% loaded ZnO nanocomposite in Table 4 but in Figure 8, it is showing that the young’s modulus value of 2 wt% loaded ZnO nanocomposite has decreased compared to 1 wt% loaded ZnO nanocomposites. Kindly correct it.
Authors: Thank you for your pointing this out. When we have carefully re-checked an Excel plotting of selected tensile curve for PLLA-PEG-PLLA /1 %wt ZnO film, we found a column selection error. We have revised this tensile curve as shown in Figure 9 on P. 12 of the revised manuscript. The young’s modulus value of 2 wt% loaded ZnO nanocomposite has slightly increased compared to 1 wt% loaded ZnO nanocomposites as follows.
Figure 9. Selected stress-strain curves of pure PLLA-PEG-PLLA and PLLA-PEG-PLLA/ZnO nanocomposite films with various contents of ZnO.
Question 8. As you have mentioned in Table 4 that the opacity value of 4 wt% loaded ZnO nanocomposite is higher but in Figure 9, it is showing that 3 wt% loaded ZnO nanocomposite is more opaque than 4 wt% loaded ZnO nanocomposite. Kindly correct it.
Authors: Photographs of film samples have been coorected according to your suggestion as shown in Figure 10 on P. 13 of the revised manuscript as follows.
Figure 10. Photograph of (a) pure PLLA-PEG-PLLA film and PLLA-PEG-PLLA/ZnO nanocomposite films with nano-ZnO contents of (b) 1 %wt, (c) 2 %wt, (d) 3 %wt, and (e) 4 %wt.
Question 9. In conclusion section, authors need to mention the novelty of their work.
Authors: The novelty of this work has been described in conclusion section on P. 16, lines 503-507 of the revised manuscript as follows.
It can be concluded that the nano-ZnO shows promise as a multi-functional filler to improve the crystallization, mechanical, UV-barrier, and antibacterial ability of the flexible PLLA-PEG-PLLA. The PLLA-PEG-PLLA/ZnO nanocomposites have the potential for use as novel flexible, biodegradable and antibacterial food packaging.
Question 10. Authors need to check the English language throughout the manuscript and need to correct it.
Authors: The revised manuscript has checked on English language by an English native speaker.
Concluding Remarks
We hope that our responses answer the reviewers’ comments to their satisfaction and that the revisions that have been made to the paper enhance its clarity for the benefit of the reader.
Yours Faithfully,
The Authors

Round 2
Reviewer 1 Report
Comments and Suggestions for Authors
Thank you for your responses to my comments.
I now understand your point of view.
I wish you success
Reviewer 2 Report
Comments and Suggestions for Authors
The authors have modified the article as per my suggestions. Hence, this article now can be accepted in its present form.